# Platelet Lysate Therapy Attenuates Hypoxia Induced Apoptosis in Human Uroepithelial SV-HUC-1 Cells through Regulating the Oxidative Stress and Mitochondrial-Mediated Intrinsic Apoptotic Pathway

**DOI:** 10.3390/biomedicines11030935

**Published:** 2023-03-17

**Authors:** Zong-Sheng Wu, Hou-Lun Luo, Yao-Chi Chuang, Wei-Chia Lee, Hung-Jen Wang, Michael B. Chancellor

**Affiliations:** 1Department of Urology, Kaohsiung Chang Gung Memorial Hospital and Chang Gung University College of Medicine, Kaohsiung 833, Taiwan; 2Center for Shockwave Medicine and Tissue Engineering, Kaohsiung Chang Gung Memorial Hospital and Chang Gung University College of Medicine, Kaohsiung 833, Taiwan; 3School of Medicine, College of Medicine, National Sun Yat-sen University, Kaohsiung 833, Taiwan; 4Beaumont Health System, William Beaumont School of Medicine, Oakland University, Royal Oak, MI 48073, USA

**Keywords:** hypoxia, oxidative stress, platelet-rich plasma, apoptosis

## Abstract

(1) Background: Ischemia/hypoxia plays an important role in interstitial cystitis/bladder pain syndrome (IC/BPS). Platelet-rich plasma (PRP) has been shown to relieve symptoms of IC/BPS by regulating new inflammatory processes and promoting tissue repair. However, the mechanism of action of PRP on the IC/BPS bladder remains unclear. We hypothesize that PRP might protect the urothelium during ischemia/hypoxia by decreasing apoptosis. (2) Methods: SV-HUC-1 cells were cultured under hypoxia for 3 h and treated with or without 2% PLTGold^®^ human platelet lysate (PL). Cell viability assays using trypan blue cell counts were examined. Molecules involved in the mitochondrial-mediated intrinsic apoptosis pathway, HIF1α, and PCNA were assessed by Western blot analysis. The detection of apoptotic cells and CM-H2DCFDA, an indicator of reactive oxygen species (ROS) in cells, was analyzed by flow cytometry. (3) Results: After 3 h of hypoxia, the viability of SV-HUC-1 cells and expression of PCNA were significantly decreased, and the expression of ROS, HIF1α, Bax, cytochrome c, caspase 3, and early apoptosis rate were significantly increased, all of which were attenuated by PL treatment. The addition of the antioxidant N-acetyl-L-cysteine (NAC) suppressed the levels of ROS induced by hypoxia, leading to inhibition of late apoptosis. (4) Conclusions: PL treatment could potentially protect the urothelium from apoptosis during ischemia/hypoxia by a mechanism that modulates the expression of HIF1α, the mitochondria-mediated intrinsic apoptotic pathway, and reduces ROS.

## 1. Introduction

Interstitial cystitis/bladder pain syndrome (IC/BPS) is a chronic and debilitating condition of the bladder characterized by unpleasant sensations (pain, pressure, discomfort) of the urinary bladder, associated with storage urinary symptoms [1]. There are different causes resulting in the common symptoms of IC/BPS [2]. Denudation or thinning of the urothelium, associated with infiltration of macrophages, eosinophils, or mast cells, is a common histopathological finding of the human IC/BPS bladder [3]. An increase in apoptotic cells, a decrease in proliferative cells, and elevation of apoptotic signaling molecules, including Bax, cleaved caspase-3, and Bad, have been shown in the IC/BPS bladder tissues [4]. Previous studies have reported a decrease in blood perfusion [5] and an increase in hypoxia-inducible factor-1α (HIF-1α) and VEGF expression in the bladder urothelial layer of patients with IC/BPS [6,7], which suggests a chronic microvascular ischemic condition and hypoxia occurrence in the bladder mucosa [8]. Kullmann et al. demonstrated that dysregulation of mitochondrial function and alterations in energy metabolism increase susceptibility to reactive oxygen species (ROS) generation and apoptosis in the luminal epithelium, contributing to bladder dysfunction and pain [9]. The condition of hypoxia alters the metabolism of mitochondria and increases ROS/HIF-1α-related oxidative stress and inflammation. Under pathological conditions, mitochondria are targets for reactive oxygen and nitrogen species and are critical in controlling apoptotic cell death [10]. Taken together, bladder hypoxia, mitochondrial dysfunction, and the inflammatory process may cause urothelium apoptosis and contribute to the pathology of IC/BPS.

Platelet-rich plasma (PRP) is rich in growth factors and secretes cytokines that can modulate a new inflammatory process and promote the resolution of unhealed wounds [11,12]. In addition, platelets have recently been used in mitochondrial transplantation as donors of small-size platelet mitochondria for the treatment of ischemia/reperfusion injury (IRI) in various tissues [13].

Dönmez et al. reported that intravesical instillation of PRP increased the mitotic index in HCl and cyclophosphamide-induced cystitis models in rabbits [14]. Repeated PRP intravesical injection has been proven to relieve symptoms of IC/BPS in a human study [15,16]. Platelet lysate (PL) is a kind of platelet derivative obtained by platelet destruction through freeze–thawing of a PRP sample [17,18,19]. PL, containing a plethora of growth-promoting factors, has been considered a safe alternative to fetal bovine serum for ex vivo expansion of mesenchymal stem cells. Among the advantages of PL compared to PRP, it can be frozen and stored, and an allogenic use for PL might also be conceivable [18].

In the present study, we established an ischemia/hypoxia model in vitro using SV-HUC-1 cells to determine whether PL can protect the urothelium by regulating mitochondrial function and its possible signal transduction pathways.

## 2. Materials and Methods

PLTGold^®®^ Human Platelet Lysate (Sigma-Aldrich, St. Louis, MO, USA) is derived from normal human donor platelets collected at US blood centers. Multiple donor units are pooled in large batch sizes and manufactured to produce a consistent product [15].

### 2.1. Cell Culture

SV-HUC-1 cells (ATCC, Manassas, VA, USA), derived from the urothelium lining of a benign human ureter and immortalized with SV40, were used as an in vitro model of normal urothelium [18]. SV-HUC-1 cells were cultured in 90% Ham’s F12 medium containing 10% fetal bovine serum. Petri dishes containing cells were kept in a 37 ℃ humidified incubator with a mixture of 95% air and 5% CO_2_ and 100 units/mL penicillin and 100 µg/mL streptomycin. Before the experiments, cells were grown to ~90% confluence in culture dishes. 2 × 10^5^ SV-HUC-1 cells per well were seeded in Petri dishes and incubated for 3 days.

### 2.2. Simulated Ischemia/Hypoxia Model

Petri dishes containing cells were incubated in a hypoxic incubator chamber with 95% N_2_ and 5% CO_2_. The hypoxia incubator chamber is designed to maintain a hypoxic environment for cell culture [19]. To determine the appropriate time to induce apoptosis, the SV-HUC-1 cells were treated under the ischemia/hypoxia conditions for 1, 3, or 5 h.

### 2.3. Trypan Blue Cell Counting 

A mix of 20 µL of 0.4% trypan blue with 10 µL of cell suspension were added to the hemocytometer. Viable cells are clear and unstained under an inverted microscope; dead cells were partially or completely stained blue. Cells in the four squares of the counting card were counted using a phase-contrast microscope.

### 2.4. PLTGold^®^ Human Platelet Lysate (PL) Therapy

After 3 h of hypoxia, SV-HUC-1 cells were transferred back to the culture medium in 95% air and 5% CO_2_ for reoxygenation for 24 h and treated with or without 2% PLTGold^®^ Human Platelet Lysate (Sigma-Aldrich, St. Louis, MO, USA).

### 2.5. Western Blot Analysis for HIF-1α, PCNA, Bax, Cytochrome c, Caspase3 

The cells were homogenized in M-PER Mammalian Protein Extraction Reagent (Thermo, Waltham, MA, USA), and total protein was measured with the Bradford protein assay method (Bio-Rad Laboratories, Hercules, CA, USA). Expression of HIF-1α, PCNA, Bax, cytochrome c, caspase3, and GAPDH was analyzed according to the standard protocol (Amersham Biosciences, Woburn, MA, USA). SDS-polyacrylamide gel electrophoresis was performed using the Laemmli buffer at a constant voltage of 100 V for 1 h, then transferred to a Hybond-P PVDF Membrane (Amersham Biosciences, Woburn, MA, USA). The membrane was blocked with blocking agent and then immunoblotted overnight at 4 °C with mouse anti-GAPDH mono-clonal antibody (1:10,000 dilution; Millipore, Burlington, MA, USA), rabbit anti-HIF-1α polyclonal antibody (1:1000 dilution; Proteintech, Rosemont, IL, USA), mouse anti-PCNA polyclonal antibody (1:1000 dilution; Cell Signaling, Danvers, MA, USA), rabbit anti-Bax polyclonal antibody (1:1000 dilution; Proteintech, Rosemont, IL, USA), rabbit anti-cytochrome c polyclonal antibody (1:1000 dilution; Cell Signaling, Danvers, MA, USA), and rabbit anti-caspase3 polyclonal antibody (1:1000 dilution; Cell Signaling, Danvers, MA, USA). After washing, the membrane was incubated with a secondary antibody using 5% defatted milk powder in PBST for 1 h at room temperature using a horseradish peroxidase-linked anti-rabbit or anti-mouse immunoglobulin G. Western blots were visualized by an enhanced chemiluminescence detection system (Amersham Biosciences, Woburn, MA, USA) using GAPDH as a loading control. Quantitative analysis was performed using LabWorks Image Acquisition and Analysis software.

### 2.6. Chemical Treatments

In some experiments, cells were treated with 5 mM N-acetyl-L-cysteine (NAC) (Sigma-Aldrich, St. Louis, MO, USA) to prevent ROS accumulation. For this concentration, NAC was able to increase cell viability and de-crease ROS levels [20]. Chemical treatments were given after 3 h of hypoxia, immediately at the beginning of reoxygenation.

### 2.7. Apoptosis Detection 

After 3 h of hypoxia, SV-HUC-1 cells were transferred back to culture medium in 95% air and 5% CO_2_ for reoxygenation for 24 h and treated with or without PL or NAC. Apoptotic cells were quantified using the FITC Annexin V Apoptosis Detection Kit (BD Biosciences, Franklin Lakes, NJ, USA) according to the manufacturer’s protocols. Briefly, after treatment with PL or NAC, conditioned medium and cells were collected, resuspended in binding buffer, and incubated with Annexin V-FITC and PI for 15 min at room temperature in the dark. The stained cells were analyzed by flow cytometry within 1 h. Cells typically range from FITC Annexin V and PI negative (viable cells) to FITC Annexin V positive and PI negative (early stage apoptosis) and FITC Annexin V and PI positive (late-stage apoptosis). Data were examined using FlowJo software (version 7.6.1, Tree Star Inc., Ashland, OR, USA).

### 2.8. ROS Measurement

The culture medium was removed, and then the cells were washed with PBS. They were subsequently incubated for 20 min at 37 °C with 10 μM CM-H2DCFDA (Invitrogen, Waltham, MA, USA) to measure total cellular reactive oxygen species (ROS). The cells were washed with PBS, removed from the plates by pipetting with 1% trypsin containing 1 mM EDTA, pelleted at 500× *g* for 5 min, immediately resuspended in PBS, and analyzed by flow cytometry.

### 2.9. Statistical Analysis

All data were presented as means ±SE. Parameter values were compared using one-way ANOVA by Tukey test and paired *t*-test, with *p* < 0.05 considered significant. Statistical analysis was undertaken using SPSS v.18.0 (IBM Corp., Armonk, NY, USA).

## 3. Results

### 3.1. Ischemia/Hypoxia Reduced SV-HUC-1 Cell Viability and Increases Expression of HIF-1α

The viability of SV-HUC-1 cells was detected by the trypan blue cell counting after incubation in low oxygen conditions for 0, 1, 3 or 5 h. As shown in Figure 1, the number of viable cells gradually decreased after increasing the hypoxia time (Figure 1) (hypoxia 1 h vs. control group; ** *p* < 0.01; hypoxia 3 h vs. control group; *** *p* < 0.001; hypoxia 5 h vs. control group; *** *p* < 0.001). The affected cells displayed a typical apoptotic morphology with cell shrinkage and membrane blebbing. The death of about 50% of the cell population resulted within 3 h. Therefore, cells incubated under hypoxia conditions for 3 h were used in the subsequent experiments. The expression of HIF 1α significantly increases in the SV HUC-1 cells cultured after 1 h of hypoxic conditions and persisted at 5 h (Figure 2) (hypoxia 5 h vs. control group, * *p* < 0.05; hypoxia 1 h vs. control group; *** *p* < 0.001; hypoxia 3 h vs. control group, *** *p* < 0.001).

### 3.2. Effect of 2% PLTGold on Cell Viability and Expression of HIF-1α, PCNA, Bax, Cytochrome c and Caspase3

The number of viable cells was significantly decreased by hypoxia 3 h, which was ameliorated by PL treatment (Figure 3) (* control vs. hypoxia 3 h group; *** *p* < 0.001; + control vs. hypoxia 3 h group+2% PL; ++ *p* < 0.01; # hypoxia 3 h vs. hypoxia 3 h group+2% PL group. ## *p* < 0.01). Western blotting showed the upregulation of HIF-1α, Bax, cytochrome c, and caspase 3 and downregulation of PCNA under 3 h of hypoxia, which was ameliorated by PL treatment (Figure 4) (* control vs. hypoxia 3 h group; * *p* < 0.05; # hypoxia 3 h vs. hypoxia 3 h group+2% PL group. # *p* < 0.05).

### 3.3. Effects of 2% PLTGold on Hypoxia-Induced Apoptosis by Annexin V-FITC/PI Flow Cytometry Analyses

The percentage of the non-apoptotic (lower left quadrant), early apoptotic (lower right quadrant), and late apoptotic or necrotic cells (upper right quadrant) was assessed by Annexin V and propidium iodide staining. The results from flow cytometry analyses indicated that the early stage apoptosis rate was significantly increased after 3 h hypoxia exposure (Figure 5), which was ameliorated by PL treatment (* control vs. hypoxia 3 h group; *** *p* < 0.001; + control vs. hypoxia 3 h group+2% PL; +*p* < 0.05, +++ *p* < 0.001; # hypoxia 3 h vs. hypoxia 3 h group+2% PL group. ### *p* < 0.001).

### 3.4. Effects of 5 mM NAC on Hypoxia-Induced Apoptosis by Annexin V-FITC/PI Flow Cytometry Analyses

The results from flow cytometry analyses indicated that 5 mM NAC significantly decreased the late stage apoptosis (Figure 6) (* control vs. hypoxia 3 h group; *** *p* < 0.001; + control vs. hypoxia 3 h group+5 mM NAC; +++ *p* < 0.001; # hypoxia 3 h vs. hypoxia 3 h group+5 mM NAC group, ### *p* < 0.001).

### 3.5. Effects of 2% PLTGold on Hypoxia-Induced ROS Expression by Flow Cytometry Analyses

Intracellular ROS levels, assessed by using CM-H2DCFDA, were significantly increased in the hypoxia 3 h group and ameliorated by PL treatment (Figure 7) (* control vs. hypoxia 3 h group; ** *p* < 0.01; + control vs. hypoxia 3 h group+2% PL; + *p* < 0.05; # hypoxia 3 h vs. hypoxia 3 h group+2% PL group. # *p* < 0.05).

## 4. Discussion

IC/BPS is a chronic inflammatory disease of the bladder associated with chronic hypoxia and damage to the urothelium. SV-HUC-1 cells are often used in urothelium in vitro studies, and they have been reported to investigate the mechanisms by which ketamine induces toxicity in human urothelium [21]. The use of ischemia-hypoxia cell models in vitro has been reported in the study of Asiaticoside against ischemia-hypoxia in cultured rat cortex neurons [22], and mitochondrial-derived vesicles protect cardiomyocytes against hypoxic damage [23]. Our in vitro study using the hypoxic model of SV-HUC-1 demonstrated that hypoxia-induced apoptosis was related to the mitochondrial pathway, and PL treatment could partially reverse the changes. 

It has been suggested that ischemia/hypoxia conditions occur in the bladder mucosa and contribute to IC/BPS symptoms [5,6,7,8]. Furthermore, mitochondrial dysfunction, a major target in hypoxic/ischemic injury, has been linked to HCl-induced cystitis in rats [24]. The current study revealed that the viability of SV-HUC-1 cells decreased in association with a significant increase in reactive oxygen species and apoptosis, upregulation of HIF-1α, Bax, cytochrome c, and caspase 3 expression, and downregulation of PCNA expression after 3 h of hypoxia, which were partially reversed by PL treatment. Cells during the ischemic period may become damaged, leading to energy declines caused by hypoxia and deprivation of metabolic substrates. Mitochondrial dysfunction has long been considered to play a role in hypoxic and ischemic cell death [25]. Hypoxia induces time-dependent translocation of cytosolic Bax to mitochondrial membranes associated with mitochondrial permeability alterations, release of cytochrome c, and caspase activation, which con-tribute apoptogenic mechanisms to these pathophysiological processes [26,27].

Lee et al. demonstrated that high expression and co-localization of metallothionein and HIF-1α were found in the bladder mucosa of patients with IC/BPS [28]. Akiyama et al. has demonstrated that the upregulation of HIF-1α in the Hunner lesions of IC/BPS bladder induces an activation of HIF1α-related biological pathways and leads to the upregulation of VEGF (vascular endothelial growth factor), erythropoietin, iNOS (inducible nitric oxide synthase), and/or glucose transporters, which protect cells from lethal damage or apoptosis induced by ischemia or inflammation [7]. Our current study revealed that under hypoxic conditions, SV-HUC-1 cells induced increased expression of HIF-1α, Bax, cytochrome c, caspase 3, cell apoptosis, and decreased expression of PCNA, in which molecular changes were partially reversed by PL treatment. Our results suggest that HIF-1α could be considered a biomarker for hypoxia/ischemia in IC/BPS, and PL treatment may ameliorate the mitochondrial apoptotic pathway and normalize the level of HIF-1α.

Colgan et al. suggested that hypoxia plays a prominent role in inflammation, so-called “inflammatory hypoxia,” which results from a combination of recruited inflammatory cells and the activation of multiple O_2_-consuming enzymes during inflammation [29]. These changes in tissue oxygenation result in a hypoxic microenvironment, mitochondrial dysfunction, an increase of reactive oxygen species, and lead to oxidative damage, which further promotes inflammation [30]. Mitochondria are both the primary source and target of ROS. Oxidative damage to mtDNA could lead to mitochondrial dysfunction and, in turn, trigger the inflammatory response [31].

The response to hypoxia is mediated by HIF activation that regulates the expression of a cohort of genes that promote adaptation to hypoxia and contribute to the maintenance of epithelial barrier function, nutrient absorption, and immune regulation. However, chronic HIF activation exacerbates disease conditions, leading to epithelial injury and inflammation [32]. Taken together, HIF-1α may play a protective or detrimental effect during hypoxia/ischemia, depending on the character of the tissue and hypoxia/ischemia model. 

PRP therapy is widely used in a variety of regenerative medicine, including musculoskeletal disorder and urogenital disorders, because platelet growth factors (PGFs) modulate the three phases of the wound healing and repair cascade (inflammation, proliferation, remodeling) [33]. Soliman et al. reported that PRP ameliorates gamma radiation-induced nephrotoxicity via modulating oxidative stress and apoptosis in a rat model [34]. PL is a kind of platelet derivative obtained by platelet destruction by freeze–thawing of a PRP sample [17,18,19]. PL has been considered a PRP because it can be stored frozen, and allogenic use for PL might also be possible [18]. Our current study showed that PL could regulate hypoxia-induced early apoptosis by reducing ROS, cytochrome c, and caspase3.

NAC is a well-known antioxidant that prevents oxidative stress-induced damage to key cellular components. NAC is clinically used to support antioxidant activity under conditions of acetaminophen poisoning, stress, infection, and inflammation [30,31]. The current study revealed that NAC exerted cytoprotective effects against hypoxia-induced late apoptosis in SV-HUC-1 cells and had a minor effect on cell viability.

## 5. Limitations

The current model of in vitro SV-HUC-1 cell hypoxia and PL treatment is not a chronic ischemia study. Further in vivo studies and human clinical trials should be conducted to elucidate the role of hypoxia in IC/BPS and the potential utility of PL as a novel therapeutic treatment.

## 6. Conclusions

In conclusion, we show that platelet lysate exerts a protective effect against hypoxia-induced apoptosis in SV-HUC-1 cells by regulating oxidative stress and the mitochondrial-mediated intrinsic apoptotic pathway (Figure 8). Additionally, NAC modulating oxidative stress may also have inhibitory effects on hypoxia/ischemia-induced apoptosis. 

## Figures and Tables

**Figure 1 biomedicines-11-00935-f001:**
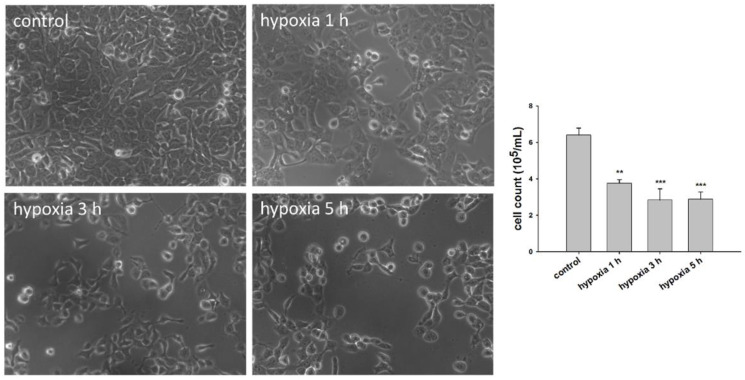
Contrast cell images of SV HUC-1 cells. Hypoxia marked decreased cell viability. (n = 5, ** *p* < 0.01, *** *p* < 0.001 vs. control group. magnification ×400).

**Figure 2 biomedicines-11-00935-f002:**
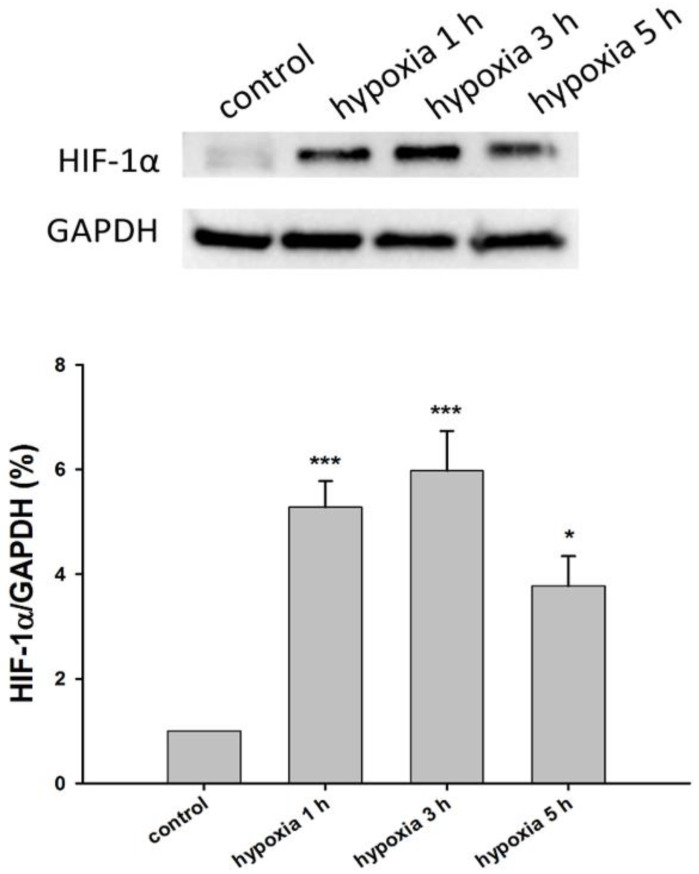
The representative immunoblotting and relative intensity of HIF-1α protein in the control and hypoxia groups. The level of HIF-1α was significantly increased at hypoxia 1, 3 and 5 h. (n = 5, * *p* < 0.05, *** *p* < 0.001 vs. control group).

**Figure 3 biomedicines-11-00935-f003:**
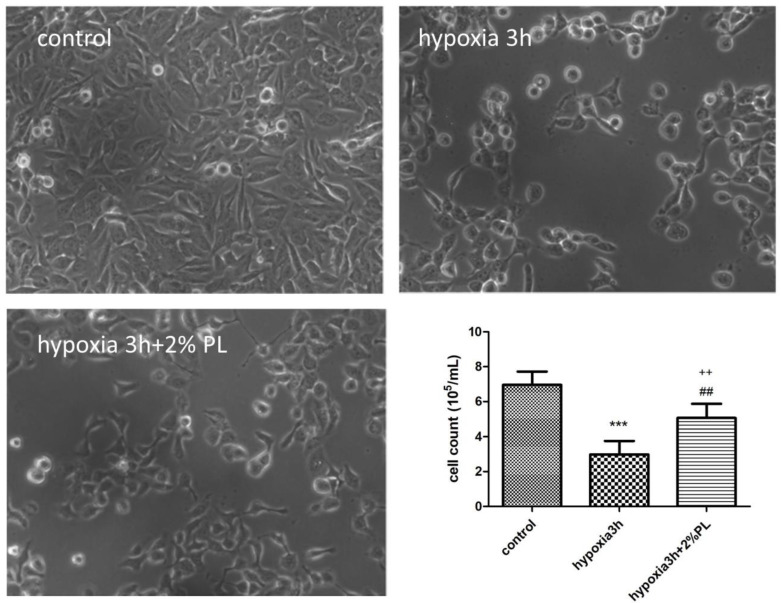
The number of SV-HUC-1 viable cells was significantly decreased by hypoxia 3 hr, which was partially reversed by 2% PL treatment. (n = 5, magnification ×400, * control vs. hypoxia 3 h group; *** *p* < 0.001; + control vs. hypoxia 3 h group+2% PL; ++ *p* < 0.01; # hypoxia 3 h vs. hypoxia 3 h group+2% PL group. ## *p* < 0.01).

**Figure 4 biomedicines-11-00935-f004:**
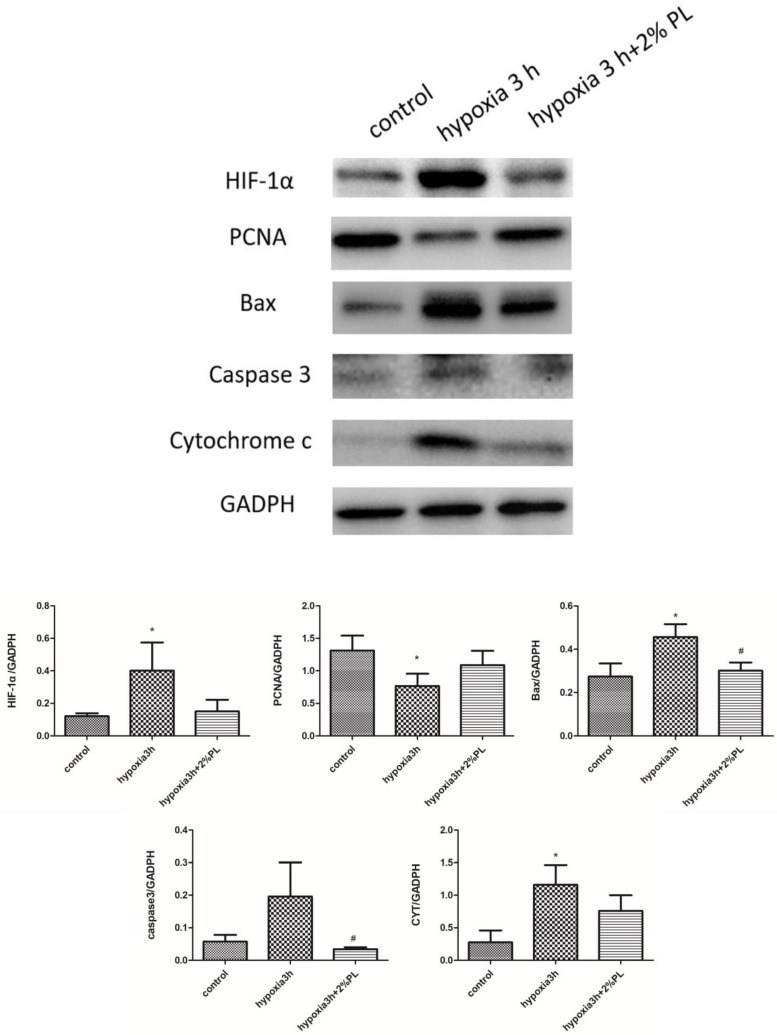
SV-HUC-1 cells were treated by hypoxia for 3 h followed by 2% PL. Western blot for HIF-1α, PCNA, Bax, cytochrome c, and caspase3 protein expression. (n = 5, * control vs. hypoxia 3 h group; * *p* < 0.05; # hypoxia 3 h vs. hypoxia 3 h group + 2% PL group. # *p* < 0.05).

**Figure 5 biomedicines-11-00935-f005:**
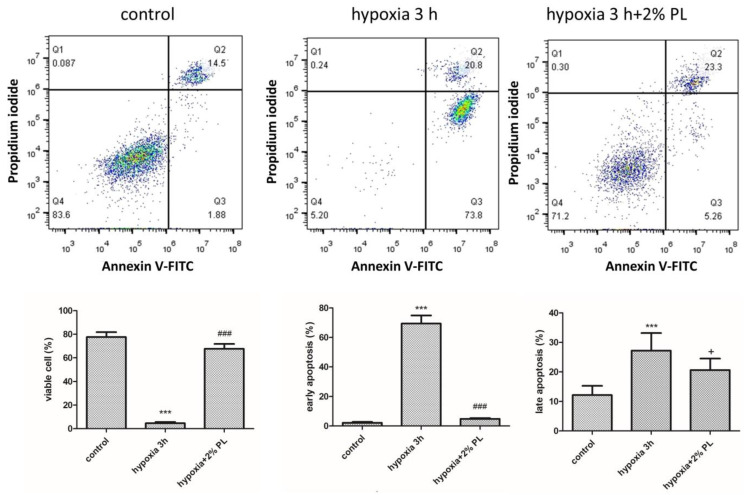
The effects of 2% PL on hypoxia-induced apoptosis by Annexin V-FITC/PI flow cytometry analyses. Non-apoptotic cells were Annexin V negative and PI-negative. Early apoptotic cells were Annexin V-positive but PI-negative. Late apoptotic cells, together with necrotic cells, were stained intensely with PI. Viable cells were significantly decreased, and early-stage apoptosis rates were significantly increased after 3 h hypoxia exposure. Additionally, 2% PL significantly increased the viable cells and decreased the early-stage apoptosis of SV-HUC-1cells. (n = 5, * control vs. hypoxia 3 h group; *** *p* < 0.001; + control vs. hypoxia 3 h group + 2% PL; + *p* < 0.05; # hypoxia 3 h vs. hypoxia 3 h group + 2% PL group. ### *p* < 0.001).

**Figure 6 biomedicines-11-00935-f006:**
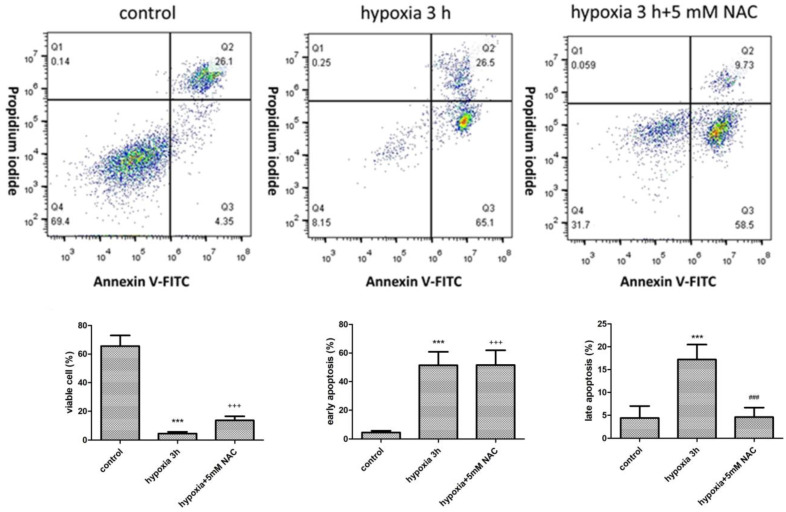
The effects of 5 mM NAC on hypoxia-induced apoptosis by Annexin V-FITC/PI flow cytometry analyses (n = 5). Non-apoptotic cells were Annexin V negative and PI-negative. Early apoptotic cells were Annexin V-positive but PI-negative. Late apoptotic cells, together with necrotic cells, were stained intensely with PI. Viable cells were significantly decreased, and early-stage apoptosis rates were significantly increased after 3 h hypoxia exposure. Additionally, 5 mM NAC significantly decreased the late-stage apoptosis of SV-HUC-1 cells. (* control vs. hypoxia 3 h group; *** *p* < 0.001; + control vs. hypoxia 3 h group+5 mM NAC; +++ *p* < 0.001; # hypoxia 3 h vs. hypoxia 3 h group+5 mM NAC group, ### *p* < 0.001).

**Figure 7 biomedicines-11-00935-f007:**
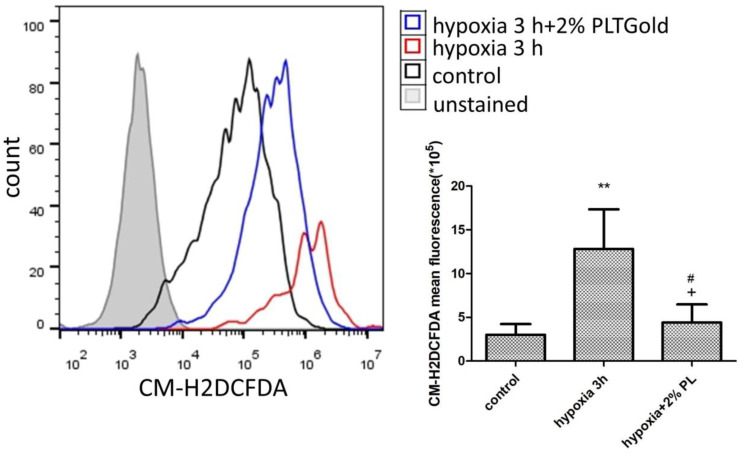
Flow cytometry of CM-H2DCFDA staining for ROS in hypoxia 3 h with or without 2% PL. (n = 5, * control vs. hypoxia 3 h group; ** *p* < 0.01; + control vs. hypoxia 3 h group + 2% PL; + *p* < 0.05; # hypoxia 3 h vs. hypoxia 3 h group + 2% PL group. # *p* < 0.05).

**Figure 8 biomedicines-11-00935-f008:**
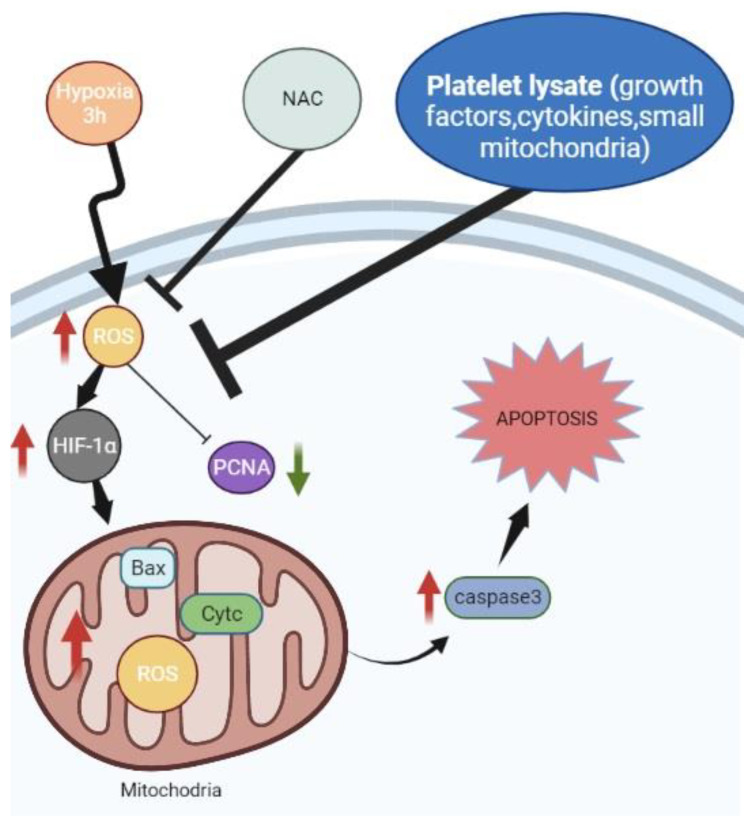
Platelet lysate inhibits hypoxia induced apoptosis by regulating the production of ROS, and expression of HIF-1α, Bax, cytochrome c and caspase 3 proteins. NAC reduces the production of ROS and inhibits apoptosis.

## Data Availability

Not applicable.

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
