# Peer review of "Platelet Lysate Therapy Attenuates Hypoxia Induced Apoptosis in Human Uroepithelial SV-HUC-1 Cells through Regulating the Oxidative Stress and Mitochondrial-Mediated Intrinsic Apoptotic Pathway"

_biomedicines, 2023, doi:10.3390/biomedicines11030935_

Round 1
Reviewer 1 Report (Previous Reviewer 1)
In the revised manuscript there are still grammar errors.
In the results section statistical significances (as P values) of comparisons between the groups must be given in text and must be attributed to respective comparisons. Here authors simply copied and pasted them from the figure legends.
Author Response
Reviewer 1
Comments and Suggestions for Authors
In the revised manuscript there are still grammar errors.
A: The Senor Author, Professor Michael Chancellor, a native English speaker, has revisited the article again and correct the grammar errors.
In the results section statistical significances (as P values) of comparisons between the groups must be given in text and must be attributed to respective comparisons. Here authors simply copied and pasted them from the figure legends.
A: We appreciated reviewer’s comment. The statistical significance and P value of the comparison between groups are added to the result section of the article.
Reviewer 2 Report (Previous Reviewer 2)
Dear Authors,
I appreciate your interest in the suggestions made, and I hope I've helped improve your article.
Author Response
Reviewer 2
Comments and Suggestions for Authors
Dear Authors,
I appreciate your interest in the suggestions made, and I hope I've helped improve your article.
A: Thank you for the improvement of the article.
Once again, we appreciate that the Biomedicines is interested in this research effort and we have incorporated all of these comments into our revised manuscript to produce a better final manuscript in the Biomedicines. We hope the revised manuscript is now acceptable for publication in the “Biomedicines”.
This manuscript is a resubmission of an earlier submission. The following is a list of the peer review reports and author responses from that submission.
Round 1
Reviewer 1 Report
In this manuscript Wu et al., focused on the acute effects of Platelet Lysate (PL) treatment of SV-HUC-1 cell apoptosis and proliferation induced by in-vitro hypoxia where they intended to model interstitial cystitis/bladder pain syndrome (IC/BPS).
IC/BPS is a chronic condition associated with chronic inflammation, apoptosis, decreased proliferation and hypoxia of the bladder tissue based on in-vivo findings. Here in this study authors aimed to explore the underlying mechanisms of apoptosis by hypoxia.
They exposed SV-HUC-1 cells to acute hypoxia for 3h and applied PL for 24h in normoxic conditions. They measured the protein expression of common apoptosis markers, cell proliferation and ROS. They found that 24h treatment of PL reversed increased apoptosis, decreased cell proliferation and ROS by acute hypoxia. Concerns about the study has been given below.
Comments:
IC/BPS is a chronic inflammatory disease of bladder associated with chronic hypoxia and damage of urothelium. However, experiments were only performed in acute hypoxia exposed cells. This is not a valid model for the intention of the study.
It is stated that SV-HUC-1 cells were transferred back to culture medium in 95% air and 5% CO2 for reoxygenation of 24 h and treated with or without 2% PLTGold® Human 95 Platelet Lysate (PL). In this setting authors are studiyng the effect of hypoxia/reoxygenation along with PL. Did the authors also add PL before hypoxia exposure?
How did the authors normalize apoptosis experiments?
Did the authors measure ROS accumulation in hypoxic conditions? From the experiments it sounds like they incubated the cells in normoxia which makes sense because the ROS levels then is not increased in hypoxia but hyp/reoxy. Same also applies to apoptosis experiments.
Trypan blue cell counting section should be shortened and carefully written.
How did the authors test 5mM NAC toxicity? It is not clear when NAC is given.
The details of early-late apoptosis measurements must be given in methods section.
There are no bar graphs of the WB data but in legends statistical significances are given.
Statistical significance signs must be simplified in most of the figures and must be given in the result section.
There are gramer and punctation errors.
Reviewer 2 Report
For authors,
The authors have conducted a novel and interesting research about the potential of Platelet Lysate Therapy Attenuates Hypoxia Induced Apoptosis in human uroepithelial SV-HUC-1 cells through Regulating the oxidative stress and Mitochondrial-Mediated Intrinsic Apoptotic Pathway.
Major recommendations:
1. For better understanding, I suggest you elaborate in the introduction the role of mitochondria that generate reactive oxidative species (ROS) and mediate cell apoptosis and thus serve as the primary subcellular target of hypoxia/ischemia and the role of platelets that contain small mitochondria and consequently can be used as mitochondrial donor cells, the role of growth factors and secreted cytokines in modulating the inflammatory process. Possibly an additional figure.
2. Attention to all abbreviations and their meanings as they should be inserted as they first appear in text; check the manuscript.
3. I suggest to separate your Conclusions and Study limitation in different sections, for a better overview of your research.
4. Please check if the resolution of figures are according to Journal Guidelines (sufficiently high resolution of minimum 1000 pixels width/height, or a resolution of 300 dpi or higher).
Sincerely yours,